# Life Cycle Analysis of a Green Solvothermal Synthesis of LFP Nanoplates for Enhanced LIBs in Chile

**DOI:** 10.3390/nano13091486

**Published:** 2023-04-27

**Authors:** Patricio Cofré, María de Lucia Viton, Svetlana Ushak, Mario Grágeda

**Affiliations:** Departamento de Ingeniería Química y Procesos de Minerales and Center for Advanced Study of Lithium and Industrial Minerals (CELiMIN), Universidad de Antofagasta, Campus Coloso, Av. Universidad de Antofagasta, Antofagasta 02800, Chile; patricio.cofre.tapia@ua.cl (P.C.); maria.viton.albeta@ua.cl (M.d.L.V.); svetlana.ushak@uantof.cl (S.U.)

**Keywords:** batteries, LiFePO_4_, life cycle analysis, solvothermal synthesis

## Abstract

Despite the structural and electrochemical advantages of LiFePO_4_ (LFP) as a cathode material, the solid-state reaction commonly used as a method to produce it at the industrial level has known disadvantages associated with high energy and fossil fuel consumption. On the other hand, solution-based synthesis methods present a more efficient way to produce LFP and have advantages such as controlled crystal growth, homogeneous morphology, and better control of pollutant emissions because the reaction occurs within a closed system. From an environmental point of view, different impacts associated with each synthesis method have not been studied extensively. The use of less polluting precursors during synthesis, as well as efficient use of energy and water, can provide new insights into the advantages of each cathode material for more environmentally friendly batteries. In this work, a solvothermal method is compared to a solid-state synthesis method commonly used to elaborate LFPs at the commercial level in order to evaluate differences in the environmental impacts of both processes. The solvothermal method used was developed considering the reutilization of solvent, water reflux, and a low thermal treatment to reduce pollutant emissions. As a result, a single high crystallinity olivine phase LFP was successfully synthesized. The use of ethylene glycol (EG) as a reaction medium enabled the formation of crystalline LFP at a low temperature (600 °C) with a nano-plate-like shape. The developed synthesis method was evaluated using life cycle analysis (LCA) to compare its environmental impact against the conventional production method. LCA demonstrated that the alternative green synthesis process represents 60% and 45% of the Resource Depletion impact category (water and fossil fuels, respectively) of the conventional method. At the same time, in the Climate change and Particular matter impact categories, the values correspond to 49 and 38% of the conventional method, respectively.

## 1. Introduction

There is currently an ongoing effort on the part of governments to reduce environmental impacts caused by automobiles and other means of transportation. In this regard, increasing the insertion of electric vehicles (EV) into the vehicle fleets of many countries has helped to reduce emissions associated with fossil fuel consumption and particulate matter (PM) emissions. The development of lithium-ion batteries has played a major role in this reduction because it has allowed the substitution of fossil fuels by electric energy as a fuel source [1]. In addition, there are international agreements for the development of better EVs [2], reduction in greenhouse gas (GWG) emissions, and increase in renewable energies in the energy mix of many countries, which together seek to reduce GWG emissions and their effect on human health and the environment [3].

Lithium-ion batteries’ electrochemical properties and performance depend mostly on the chemistry used in their electrodes, mainly in the cathode. There are several options for use as cathode material and each of them has different advantages and disadvantages, which are the subject of several studies [4,5,6]. The choice of cathode material depends almost always on economic factors, but also on the desired characteristics of the final battery, such as working voltage, lifetime, and safety [7]. For these reasons, some authors have been working to further develop the commercial production process using solid-state synthesis, as it is easy to scale up to industrial dimensions and simultaneously maintains a good electrochemical performance (135 mAhg^−1^ at a rate of 5C) [8].

One group of materials that has interested researchers and the industry owing to its electrochemical capabilities and production advantages at the industrial level is the set of olivine-structured compounds of LiMPO_4_ (M = Fe, Mn, Co, or Ni) form, among which LiFePO_4_ (LFP) stands out. First presented by J. B. Goodenough [9], LFP has a high working voltage of 3.2V, a specific capacitance of 170 mAhg^−1^, and good structural stability, providing a lifetime of more than 2000 cycles. Additionally, the crystalline structure stability of LFP makes it thermally stable [10,11], thanks to the position of lithium atoms in its crystal structure. These are housed in octahedral channels on the b-axis in the [010] direction, which gives them freedom of movement and helps the rest of the crystalline structure to remain stable during the intercalation processes [12,13]. These additional advantages have led LFP to be considered as a key element to drive large-scale electromobility, where safety, lifetime, and production costs are key issues.

Despite its positive characteristics, LFP possesses low electronic conductivity and ionic diffusivity (which ranges from 10^−13^ and 10^−16^ cm^2^ s^−1^), which has limited its widespread use in EVs and in turn has impelled numerous studies to solve these problems [14].

Despite current efforts to improve LFP and other active materials’ performance, studies assessing the environmental impact associated with the production of these compounds are very scarce and have been limited to studying batteries’ assembly or recycling and reuse [15,16]. Other authors have focused on designing novel ways to extract raw materials such as Li_2_CO_3_ and LiOH, which are commonly used in cathode materials’ synthesis, developing more efficient and environmentally friendly methodologies than those used from brine or spodumene ore [17,18,19,20].

Generally, environmental impact studies of batteries have focused on the determination of their recyclability, or second life, based on their electrode chemistry and recovery of economically valuable materials, such as gold, copper, or rare earths. Because of this lack of general interest in the environmental impacts associated with production processes, a few authors have tried to address the problem of the synthesis of cathode material from a more environmentally friendly point of view.

In order to develop a less polluting synthesis method, Liu [21] conducted a study focused on solvent refluxing and sintering temperature optimization to reduce emissions and energy consumption for LFP production. Liu obtained positive results, obtaining an LFP that achieved a high specific capacity (161 mAhg^−1^ at rate of C/10) and a homogeneous grain size (100 to 200 nm).

Yang [22] proposed a green hydrothermal synthesis method focused on optimizing system temperature such that it could take advantage of the benefits of a supercritical fluid while being energy efficient. Furthermore, because, in wet synthesis methods, the lithium source is used in excess to ensure LFP reaction and as a means of pH control, only one-third of the dissolved lithium is used in the synthesis reaction and the rest is discarded in the used solvent. Yang reused the lithium-rich solution to produce LFP with a good electrochemical performance, reaching 167 mAhg^−1^ at a rate of C/10.

Although both researchers performed an extensive electrochemical characterization, these studies did not consider an analysis to evaluate reagent consumption and emissions produced by proposed methods, thus their environmental impacts during active material production were not determined. This lack of consideration for the environmental aspect that many “green” processes have during their studies needs to be addressed if better lithium batteries are to be developed in the future, as the projected annual sales of electric vehicles (EVs) are expected to exceed 40 million vehicles by 2028, with a required investment of $10 billion–$12 billion over the next decade from the global lithium industry [23].

In this work, we have focused on comparing two methods of active material production by means of LCA to determine their environmental impact differences with a special focus on water resource consumption, fossil fuels, and GWG emissions.

Life cycle analysis (LCA) is an environmental assessment method widely used by researchers to determine opportunities for improving production processes from an environmental point of view throughout their life cycle, from raw material extraction to final disposal. This type of study includes an inventory of all resources and emissions related to the target product, so it is also frequently used for economic studies and technical evaluation of novel processes [24].

One of the main difficulties in performing a proper LCA is to obtain information related to lithium production processes, either for economic or political reasons, because many countries have classified lithium as a strategic resource [25]. Considering those restrictions, we have conducted the study using the Environmental Footprints v3.0 database, taking as a point of comparison commercial LFP production in China studied by Xie [26] versus the low energy solvothermal method proposed by us, which includes a reflux system and a solvent step. For evaluation purposes, the study is located in Chile and based on the work of Liu and Yang, which includes lithium-rich solvent reuse, water recirculation, and better temperature control. An environmental impact assessment is performed by means of LCA to identify and quantify main raw material consumption, emission, and environmental impact.

## 2. Materials and Methods

### 2.1. LiFePO_4_/C Synthesis

To develop the high energy LFP nanoplates, a green solvothermal method was used with LiOH·H_2_O, FeSO_4_·7H_2_O, and H_3_PO_4_ as precursor materials in a 3:1:1 molar ratio and EG as a solvent. First, H_3_PO_4_ and FeSO_4_·7H_2_O were dissolved in EG to form a solution (solution A). Then, LiOH·H_2_O was dissolved in EG under stirring to obtain an Li^+^-rich solution (solution B). solution B was slowly added to solution A while being stirred vigorously at 70 °C in an N_2_ atmosphere for 45 min, as shown in Figure 1. A volume of 50 mL of distilled water was added to the A + B solution to reduce it viscosity and to induce the correct dissolution of the precursors owing the low solvation capacity. The final solution was placed in an autoclave container and heated at 180 °C for 8 h. As a product, a green precipitate was obtained, which was washed with deionized water and filtered. Through this filtration step, it is possible to separate the solvent rich in Li, but the use of a small volume of distilled water is needed to do it properly. This extra water consumption was taken in consideration in the LCA. The remaining water in the filtered LFP powder was eliminated in a drying step at 50 °C.

To perform the carbon coating, the synthesized LFP was suspended in a volume of EG and glucose in a molar ratio of 4:1, respectively. To obtain a homogeneous morphology, the mixture was stirred vigorously while being subjected to an ultrasonic treatment for 30 min and 20 kHz with an ultrasonic probe (Q700 Sonicator, Misonix, Farmingdale, NY, USA). The mixture was then calcined in a tube furnace (Shimaden SRS10A, SHIMADEN, Tokyo, Japan) at 600 °C for 6 h with a heating rate of 5 Kmin^−1^ under a protective atmosphere of N_2_/H_2_. After cooling, the powder was treated with a grinding process using a planetary mill (Fritsch pulverisette 7) for 5 min at 800 rpm to obtain a fine and homogeneous LiFePO_4_/C (LFPC) powder.

X-ray powder diffraction (XRPD) with a Bruker X-ray diffractometer with Cu K irradiation (1.5406 Å) at 40 kV and 30 mA was used to identify synthesis crystalline phases and impurities. To study LFPC particles’ size, dispersion, and the carbon content of the coating, a field emission scanning electron microscope FE-SEM, Hitachi SU5000, operating at 15 kV with an EDX module (Bruker XFlash 6I30, Billerica, MA, USA), was used.

### 2.2. Electrochemical Characterization

For electrochemical characterization, a slurry was prepared from a mixture of LFPC, conductive additive (TIMCAL C45, Xiamen Top New Energy, Xiamen, China), and polyvinylidene fluoride (PVdF powder, Xiamen Top New Energy) as binder solution, in a 80:10:10 ratio, respectively. The slurry was mixed for 5 min at 1800 rpm in a high energy mixer (Conditioning Mixter AR100) to improve its mono dispersity.

The slurry was spread on an aluminum foil using a doctor blade with an initial thickness of 120 µm and dried in an oven at 80 °C overnight to obtain a dry cathode. A calendering step was used to homogenize electrode surface and control its porosity to 34%, which was calculated according to Equation (1):(1)$$\emptyset =1-\frac{Vw}{Vd}$$
where *Vw* is the volume of the wet cathode coating and *Vd* is the volume of the dry cathode coating. Finally, the cathodes were cut and sized for use in CR2032 button cells for the electrochemical characterization.

Cells were assembled in a glove box (MBraun LabStar, Munich, Germany) using a polypropylene porous polymeric separator (Celgard 2325, Celgard, Charlotte, NC, USA); lithium hexafluoride (LiPF_6_) 1M in a solution of ethylene carbonate, diethyl carbonate, and methyl ethyl carbonate in a 1:1:1 volume ratio as electrolyte; and the prepared LFPC as a cathode (with a load of 3.38 mg of LFP per cm^2^ of electrode). Cell activation was carried out at rate of C/20, while operation cycles were performed at rates of C/10, C/5, and C/2, all at voltages of 2.5 V–3.6 V (V vs. Li^+^/Li).

Electrochemical impedance spectroscopy (EIS) was used to characterize the lithium insertion reaction and to better understand the electrochemical performance by determining the diffusion coefficient. EIS was performed with a potentiostat/galvanostat over a frequency range of 0.01 Hz–1 kHz at room temperature with a signal of 10 mVs^−1^ for 0% state of charge (SOC0) at 2.5 V after five charge–discharge cycles.

### 2.3. LCA

In order to correctly assess the impact of the processes under study, LCA must consider as a starting point the extraction of raw materials, transportation, processing site, energy mix, and particularly the consumption of important resources such as water and fossil fuels. To carry out a complete manufacturing process study, a mid-point approach method (environmental footprint) is used.

The model consists of a cradle-to-gate approach, because the manufacturing process is where the largest differences in energy and raw material consumption are produced. LCA is carried out to determine environmental impact differences between the proposed solvothermal synthesis method and commercial synthesis method studied by Xie based on solid-state synthesis.

#### 2.3.1. Scope and Function Unit

As mentioned above, the objective of this LCA is to determine differences in raw material consumption and the main environmental impacts associated with the two LFP production methods with a cradle-to-gate scope, from raw material procurement needed to manufacture each precursor until the product is obtained and ready to be moved out of factory. For this reason, LCA must consider extraction, pre-processing, synthesis, heat treatment, and post-processing stages. Production stages and limits of both systems for each synthesis method studied are detailed in Figure 2.

In the proposed model, different types of energy and resources were considered as inputs to the system, with water, fossil fuels, electricity, and solvents being among the most important. For some of these elements, recirculation or re-use points were considered, defined from experimental data. For emissions, water, solvents, GWGs, and particulate matter were considered. In most cases, emissions were calculated based on our experimental data, while in some cases (e.g., PM), data from other sources were used [26].

Figure 3 shows the conducted LCA system boundaries, including primary material extraction (e.g., LiOH and Li_2_CO_3_), solvents, and other reagents according to each method.

#### 2.3.2. Life Cycle Impact Assessment Methodology

This study was conducted using open-source software openLCA 1.10.2. Environmental impacts were calculated per 1 kg of LFPC produced considering the raw extraction and energy consumed for equipment and process, as reported by Xie and our experimental data.

For quantifying environmental impacts, the Environmental Footprint (Mid-Point) 3.0 methodology is used [27]. Four impact categories were studied, applying the following indicators: resource depletion—water, climate change, resource depletion—fossils fuels, and particulate matter/respiratory inorganics. These categories were chosen because of the importance of water as a key resource, especially in developing countries, while particulate matter, climate change, and fossil fuel consumption are associated with energy matrices in developed countries and their mining activities [28].

## 3. Results and Discussion

### 3.1. Chemical and Morphological Characterization

The synthetized LFPC XRD pattern (Figure 4) reported an absence of parasitic peaks, and all peaks could be indexed in the orthorhombic structure with space group Pnma. As expected, peaks corresponding to carbon coating were not detected because of their amorphous state. Owing to the high intensity of reported peaks, it could be assumed that synthesized LFPC powder possesses high crystallinity and requires a low temperature sintering process to improve its crystalline structure.

Scanning electron microscopy (SEM) and energy-dispersive X-ray (EDX)-synthesized LFPC powder images are shown in Figure 4. The results show that powder is mostly composed of homogeneous nano-sized primary particles with an approximate size of 40 nm (width), 180 nm (length), and 35 nm (thickness) and a carbon content of 4.8%. The particles were reported to have a plate-like shape, as expected from using EG as a solvent [29]. Despite this, the observed particles were agglomerated in an undesired way, forming a greater structure (secondary particles) (Figure 5a), reaching a size between 336 (width) and 900 nm (length). This type of agglomeration is related to the transition metal ions’ concentration in the A + B solution (e.g., Fe), which affects crystal growth during the synthesis. This growth is directly proportional to the Fe ions’ concentration, causing slower growth at the beginning of the reaction, when the ions are free, while it accelerates over time as LFP is formed, causing primary particles’ agglomeration into larger secondary particles [30]. This agglomeration could mean that the reaction time might still be reduced to limit secondary particle formation while reducing energy consumption.

LFP composition is examined by energy-dispersive X-ray (EDX), as shown in Figure 5b. Element distribution mapping indicates that Fe (green) and P (blue) are uniformly distributed and do not display any phase separation.

### 3.2. Electrochemical Characterization

LFPC electrochemical performance during charge and discharge cycles is shown in Figure 6a. The material reported an initial specific discharge capacity of 118.83 mAh^−1^ at a rate of C/10, with a coulombic efficiency of 98.7% (Figure 6a). The material performance was evaluated at different discharge rates (Figure 6b), where specific capacity varied significantly over the cycles: 118.83 mAh^−1^ (C/20); 92.64 mAh^−1^ (C/10); 62.93 mAh^−1^ (C/5); and 43.65 mAh^−1^ (C/2). Despite the above, the material showed great stability and was able to recover its original capacity after cycling at different rates, with an efficiency of 96.1%, as shown in Figure 7. This stable behavior was observed during an extended number of cycles, as shown in Figure 8, where a minimum specific capacity was lost and an average efficiency of 98.1% was reported.

As the proposed synthesis is based on the use of EG as a solvent and low temperature heat treatment, determining LPFC ionic diffusion is key to validate its electrochemical performance [31]. The electrochemical impedance spectrum reported in Figure 9 is divided into four regions: an initial high frequency point attributed to electrolyte ohmic resistance (Re), a second zone limited by a semicircle corresponding to medium-high frequency zone related to the solid electrolyte interface, a third medium frequency zone (Rct) corresponding to charge transfer, and finally a sloping straight line in the lower frequency zone corresponding to lithium ions’ diffusion in active material (Warburg impedance).

An equivalent circuit was designed with synthesized LFPC experimental data to determine Re, Rsei, Rct, and Rw values. As diffusion effect becomes more pronounced at low frequencies, the Nyquist plot shows a semicircle derived from a smaller and well-defined charge transfer process. This is followed by a 45° straight line extending indefinitely. In this case, the diffusion coefficient can be calculated by Warburg impedance (Zw) analysis, using Equation (2) [32]:(2)$$D_{Li^{+}}=\frac{R^{2}T^{2}}{2A^{2}n^{4}F^{4}C_{Li}^{2}\sigma^{2}}$$

Because smaller particles have a smaller diffusive coefficient, they perform better electrochemically, as shown in the extension of the Rct corresponding half-circle [33]. This is related to what is observed in Figure 5a, corresponding to EDX analysis, where an agglomeration of primary particles is appreciated and could be the focus of evaluation in future studies, both to improve electrochemical performance and to reduce synthesis process energy consumption. Despite the formation of secondary particles, the results of the equivalent circuit model reported in Table 1 are similar to those presented by other authors [34]. These authors reported an Rct value of near 30 ohms for particles with a size of 1.407 μm, which are larger than those reported by us. These larger particles have a higher diffusion coefficient (σ = 6.01 × 10^−14^ cm^2^s^−1^) than those presented in this study, thus demonstrating the relationship between particle agglomeration and the electrochemical performance.

### 3.3. Life Cycle Assessment

LCA allowed a comparison of both processes through standardized indicators of environmental impacts. As shown in Figure 10, of the four categories selected for analysis, those with highest impact were resource depletion—water and climate change. With respect to the resource depletion—water impact category, the process presented by Xie reports a value of 1.408 m^3^ depriv (a term used in the Environmental Footprint methodology (Midpoint indicator) to assess the potential of water deprivation), while the solvothermal process reports a value of 0.84 m^3^ depriv, which is 60% less than the conventional process. It is also important to note that the category with the highest incidence in this study is resource depletion—water, resulting in the necessity to continue optimizing the presented method to reduce this impact, considering the importance of water consumption.

Table 2 shows the top three subprocesses for each process for obtaining LFP with the highest contribution to the impact categories studied. In the climate change impact category of both methods, the highest contribution is that of Xie 2018, with a value of 3.53 kg CO_2_ equivalents, while the solvothermal method reports a contribution of 1.74 kg CO_2_ equivalents, directly attributed to natural gas consumption of high temperature ovens and the obtaining of other precursors consumed in lesser quantities as the ethylene glycol solvent production, as has been reported by other authors [35]. However, the impact of ethylene glycol diminishes considerably when it is reutilized during the synthesis process.

In the resource depletion—fossil fuels impact category, Xie’s methodology has the highest impact, with a value of 70.5 kg Sb (antimony) equivalents compared with 31.5 kg Sb equivalents, corresponding to the solvothermal method. This difference can be attributed to the Chinese energy matrix composition, which is largely composed of electricity generation from fossil fuels [36]. The final impact category analyzed is particulate matter/respiratory inorganics, in which values of 7.1 × 10^−7^ kg PM 2.5 equivalents and 2.7 × 10^−7^ kg PM 2.5 equivalents are reported for the conventional and solvothermal methods, respectively. The main contributions to this impact category are electricity generation from fossil fuels and iron precursor mining of cathode material, as shown in Table 2.

## 4. Conclusions

Nanoplate-shaped LiFePO_4_ crystals were prepared using a green solvothermal synthesis. The method was validated by LCA, evaluating water resource consumption, fossil fuel consumption, and pollutant emissions impact categories. Although the method allowed a single crystalline phase synthesis, nanoplates’ size was found to be larger than desired (from 40 nm to 336 nm) owing to the formation of secondary particles associated with reaction time in the reactor. Despite this result, the material has shown acceptable electrochemical performance (with a maximum of 118.83 mAh^−1^ at C/20) and managed to maintain stability throughout charging cycles even at different rates, from C/20 to C/2. An LCA was performed using experimental data, public databases, and research performed by other authors. The LCA reported that the solvothermal method has considerably lower impact indices in each of the studied environmental impact categories, among which water consumption and climate change impact (depriv 1.408 m^3^ versus 0.84 m^3^, and 3.53 kg CO_2_ versus 1.74 kg CO_2_ equivalents) stand out, showing evidence of the potential of this synthesis method as a more environmentally friendly production option.

## Figures and Tables

**Figure 1 nanomaterials-13-01486-f001:**
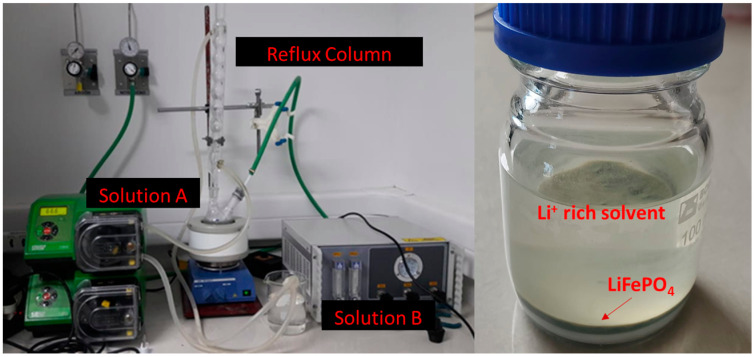
Schematic LFP synthesis reactor.

**Figure 2 nanomaterials-13-01486-f002:**
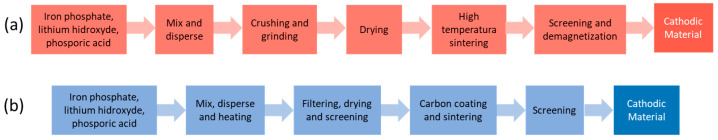
General scheme of production process stages considered for LCA. (**a**) Conventional solid- state process and (**b**) solvothermal process with recirculation.

**Figure 3 nanomaterials-13-01486-f003:**
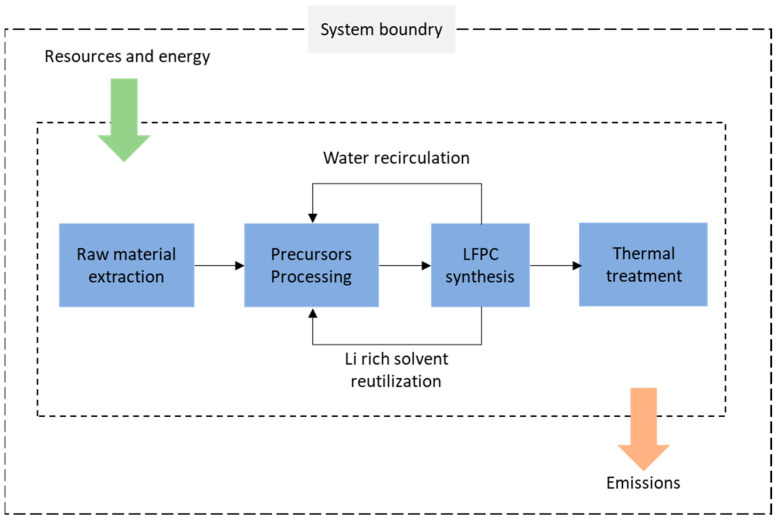
The conducted study system boundaries, including primary material extraction, production, use phase, and reutilization.

**Figure 4 nanomaterials-13-01486-f004:**
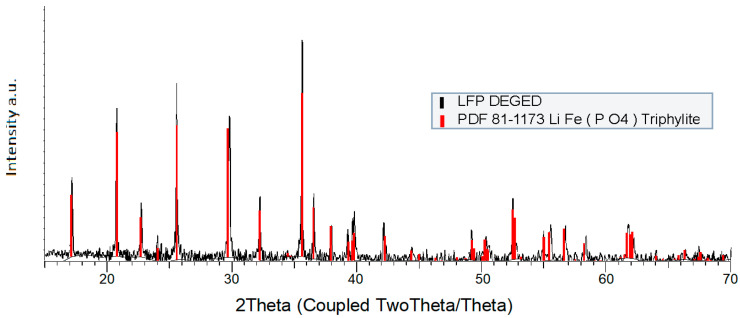
XRD patterns of synthesized LiFePO_4_ (black) compared with XRD patterns of LiFePO_4_ (red) from the database (code 04-015-6173).

**Figure 5 nanomaterials-13-01486-f005:**
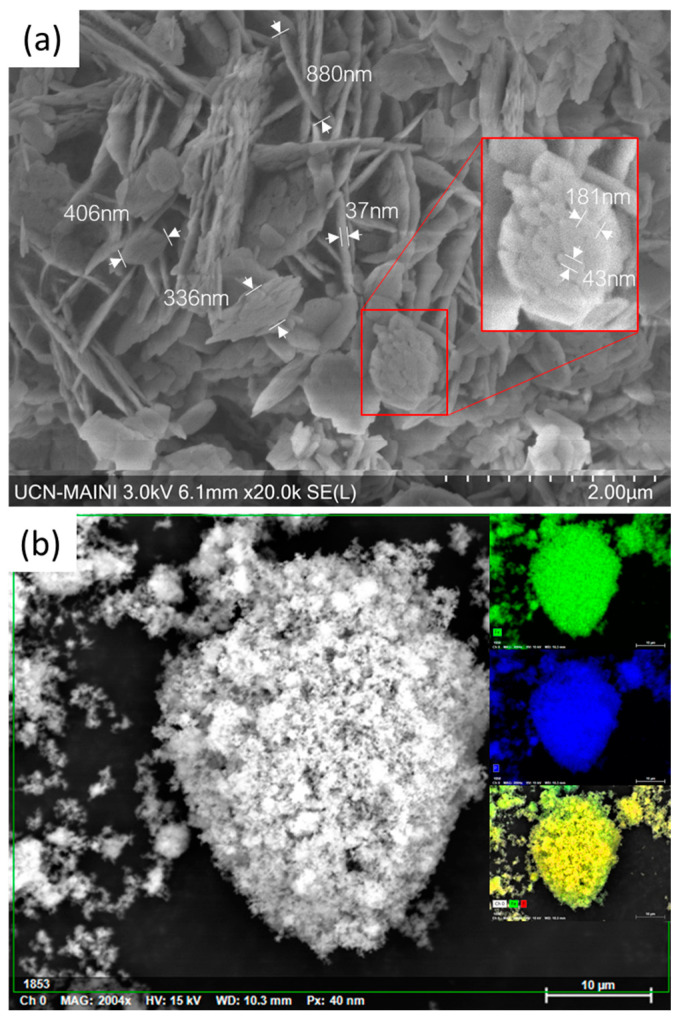
(**a**) Scanning electron microscopy (SEM) and (**b**) energy-dispersive x-ray (EDX)-synthesized LFP powders images. Miniature images correspond to sample element distribution: Fe (green), P (blue), and both (yellow).

**Figure 6 nanomaterials-13-01486-f006:**
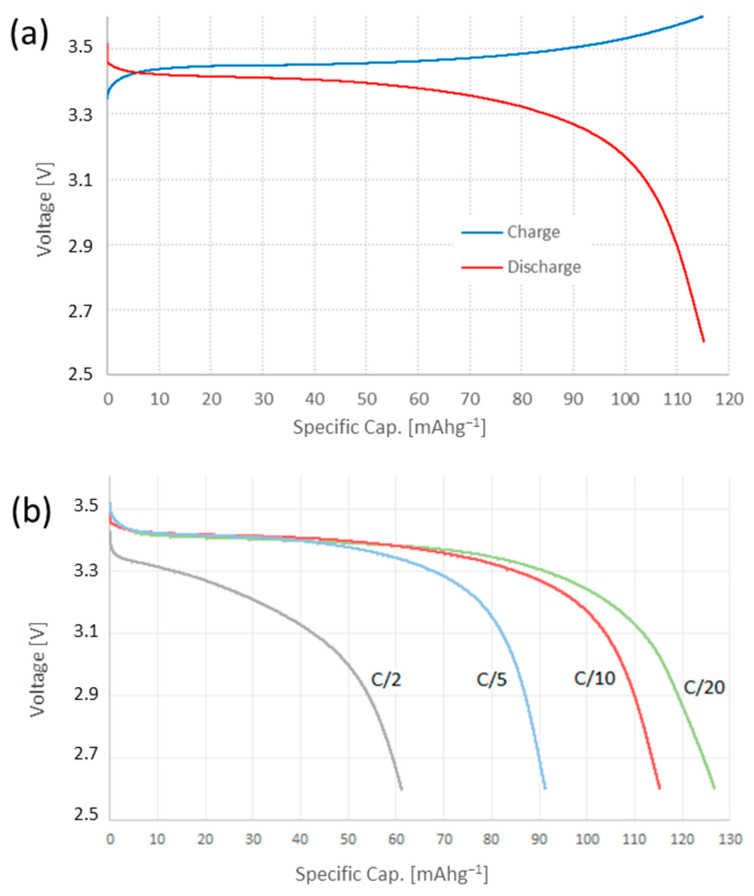
LFPC electrochemical charge/discharge performance: (**a**) coulombic efficiency and (**b**) at different rates of discharge.

**Figure 7 nanomaterials-13-01486-f007:**
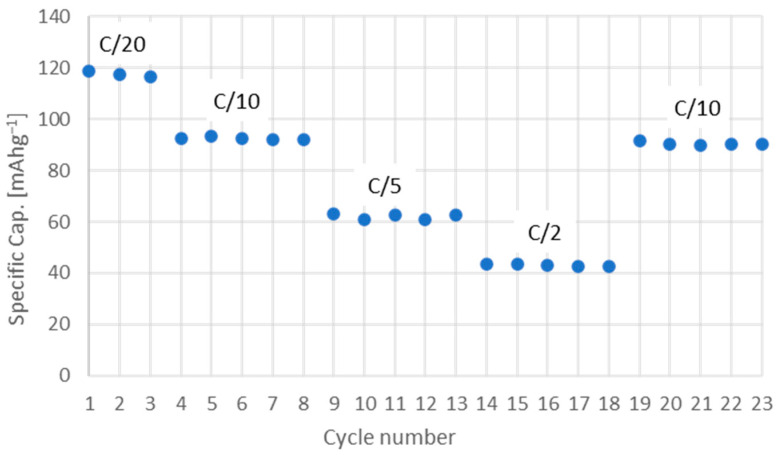
Specific discharge capacity of an LFPC cell during extended cycling at different rates. Capacity stability indicates a steady state along cycles.

**Figure 8 nanomaterials-13-01486-f008:**
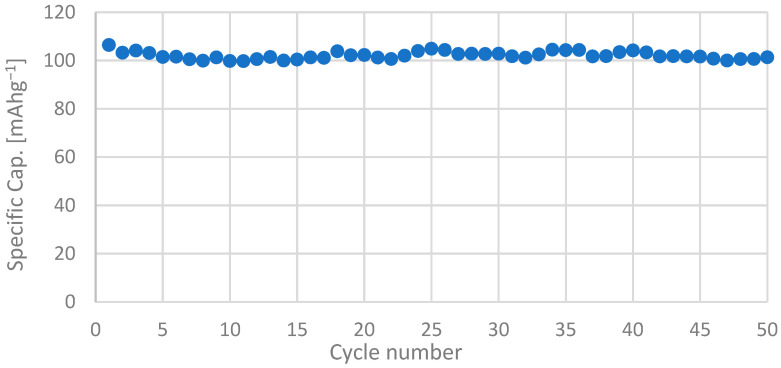
Specific discharge capacity of an LFPC cell during extended cycling at a rate of C/10. Capacity stability indicates a steady state along cycles.

**Figure 9 nanomaterials-13-01486-f009:**
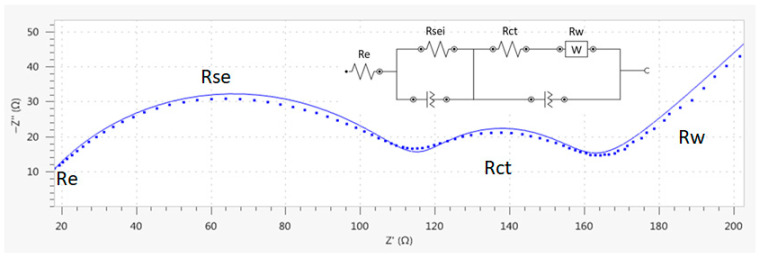
Nyquist plot for EIS measurement (SOC0) with LFPC (dots) and EIS model by equivalent circuit (line).

**Figure 10 nanomaterials-13-01486-f010:**
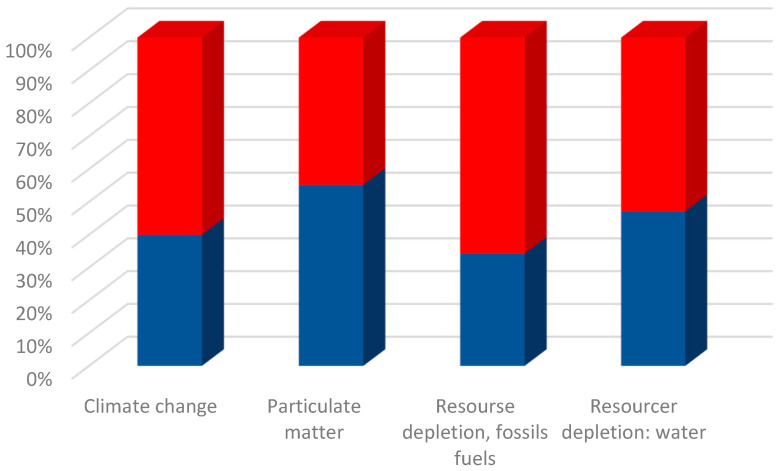
Graphic shows relative indicator results for each LFP production process, Xie 2018 (red) and solvothermal (blue). For each indicator, the maximum result is set to 100% and other process results are displayed proportionally.

**Table 1 nanomaterials-13-01486-t001:** Parameters’ values obtained by fitting the equivalent circuit model to experimental impedance spectra.

Element	Value	Unit
Re	10.31	Ω
Rse	110	Ω
Rct	37.48	Ω
RW	5.6 × 10^−2^	ohm^−1^
Ionic Diffusion (σ)	6.96 × 10^−12^	cm^2^s^−1^

**Table 2 nanomaterials-13-01486-t002:** Main contributions to this impact category.

Impact Category	Top 3 Contributions to Impact Category for LFP Solvothermal Production	Top 3 Contributions to Impact Category for LFP Conventional Production
Climate change	Ethylene glycol production (0.575 kg CO_2_ eq)	Electricity generation from hard coal (0.891 kg CO_2_ eq)
	Electricity generation from hard coal (0.369 kg CO_2_ eq)	Phosphate rock obtained at mine (0.471 kg CO_2_ eq)
	Iron (II) sulphate production (0.323 kg CO_2_ eq)	Liquid ammonia production (0.466 kg CO_2_ eq)
Particulate matter	Ethylene glycol production (1.036 × 10^−7^ kg PM 2.5 eq)	Electricity generation from hard coal (4.8 × 10^−7^ kg PM 2.5 eq)
	Iron (II) sulfate production (6.9 × 10^−8^ kg 2.5 eq)	Ferrite (iron ore) obtained at mine (6.34 × 10^−8^ kg PM 2.5 eq)
	Electricity generation from diesel (5.6 × 10^−8^ kg PM 2.5 eq)	Phosphate rock obtained at mine (3.6 × 10^−8^ kg PM 2.5 eq)
Resource depletion: fossil fuels	Ethylene glycol production (15.057 kg Sb eq)	Electricity generation from diesel (14.018 Kg Sb eq)
	Iron (II) sulfate production (6.95 kg Sb eq)	Sulphur production (9.760 kg Sb eq)
	Electricity generation from hard coal (3.72 kg Sb eq)	Electricity generation from hard coal (8.627 kg Sb eq)
Resource depletion: water	Iron (II) sulfate production (0.452 m^3^ depriv)	Liquid ammonia production (0.542 m^3^ depriv)
	Ethylene glycol production (0.231 m^3^ depriv)	Electricity generation from hydropower (0.197 m^3^ depriv)
	Phosphoric acid production (0.113 m^3^ depriv)	Soda ash production (0.164 m^3^ depriv)

## Data Availability

The data is available on reasonable request from the corresponding author.

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
