# Peer review of "Life Cycle Analysis of a Green Solvothermal Synthesis of LFP Nanoplates for Enhanced LIBs in Chile"

_nanomaterials, 2023, doi:10.3390/nano13091486_

Round 1
Reviewer 1 Report
This paper studies life cycle analysis of a green solvothermal synthesis of LFP nanoplates, which has potential application value in engineering. In order to meet the requirements of high-quality publication of the journal, it is recommended to consider the following suggestions,
1) The quantitative data for Introduction needs to be increased.
2) The innovation of this article is not reflected in the first section and needs to be modified.
3) The second Section needs to add pictures of the experimental device.
4) What is the basis for selecting the parameter in Tables 1?
5) The mehtod proposed in this paper needs to be compared with the previous literature, otherwise it cannot reflect innovation.
6) The Discussion Section needs a separate section.
7) The conclusion is too wordy and needs to be simplified.
8) There are few references in the last three years.
9) Does the format of all References meet the requirements of the journal?
Reviewer 2 Report
The manuscript is suggested to be accepted after the following issues are addressed.
1) The writing style of the paper is good in some sections and not so good in other, there are many formatting typos (missing subscripts and superscripts, missing spaces between numbers and units, upper and lower case letters are mixed up (PH, x-ray), etc.). In addition, the language of Sections 2 and 3 should be polished thoroughly.
2) The references are of wrong format, see MS preparation instructions https://www.mdpi.com/journal/nanomaterials/instructions/ (reference numbers should be placed in square brackets [ ], etc.). In addition, due to an incorrect reference management template, the journal title is omitted from the reference list for more than half of the articles, the reference list does not include DOI.
3) No uniformity in data presentation. In Section 2 and Figure 5 the C-rates are given as C/20, C/10, etc., and wrongly denoted as “velocities” (section 3), in Introduction and Section 3.2 C-rates are given as 0.2 C, 0.5 C, etc.
4) Sections 2 and 3. The following experimental details are missing:
- what was the carbon content in the synthesized LiFePO4/C powder?
- 10 wt% of PVDF should refer to dry PVDF, not to its 6% commercial solution in NMP
- how the porosity of the electrodes was measured?
- what solvents were used to prepare LiPF6-based electrolyte?
- how good was the long-term cycling performance of the cell, what was the capacity decay over 1000 cycles? (corresponding figure should be provided)
- at what potential was the EIS spectrum measured? How long was the cell previously cycled and in what conditions? (the cell 'biography' may affect the result greatly)
5) Another case of inconsistency and inaccuracy in the MS:
Lines 219-223 present accurate description of LFP morphology of particles with plate-like shape (sheets, platelets) “Results show powder is mostly composed of homogeneous nano-sized primary particles with an approximate size of 40nm (width) 220 and 180 nm (length) and 35nm (thickness) with a plate-like shape, as expected when using EG as a solvent (Ludwig et al., 2016). These particles are agglomerated into secondary particles sharing sheet or plate shape, as shown in Fig. 4a., reaching a size between 40nm (width) and 900 nm (length). “
But in Conclusions, Lines 323-327, the LFP particles are referred as “dish-shaped” and “nano-silver plates”. (“Dish-shaped LiFePO4 nanocrystals have been prepared by a green solvothermal synthesis. …. Although the method allowed a single crystalline phase synthesis, nano-silver plates size has been found to be larger than desired due to secondary particles formation associated with reaction time in the reactor.”)
Reviewer 3 Report
Patricio Cofré et al. reported a green solvothermal synthesis method of LFP nanoplates for lithium ion battery. Manuscript can be considered after major revision. Here are some suggestions to consider:
1. It is recommended that the heating rate during the 600°C annealing process be provided by the authors.
2. The caption for Figure 4 should be revised to clearly distinguish between subfigures (a) and (b).
3. The electrochemical performance metrics, including capacity, rate, EIS, and diffusion, should be compared to those reported in relevant literature.
4. authors provided the full name of "depriv" and explained how it is calculated.
5. While the solvothermal data in Table 2 is from literature, it would be valuable if the authors could include their own data obtained from this research to supplement the information.
Reviewer 4 Report
1) A comparison of environmental impacts of the different LFP synthesis techniques performed by authors is interesting and important in general. However, the applied methodology doesn’t seem fairly correct as the materials under comparison should have comparable functional properties (electrochemical capacity, fade rate etc.), and, hence, comparable market price. This factor is not taken into account in these considerations yet. First of all, the information on the morphology and electrochemical performance of LFP powders obtained by the solid state technique and studied by Xie et al. should be added.
2) The electrochemical capacity of LFP powders obtained by the hydrothermal method in this study is obviously lower than the maximum values obtained for these materials. However, taking into account very approximate character of the environmental calculations being the main objective of this paper, it is acceptable indeed, as it is claimed by authors in Conclusions. At the same time, in order to give full and correct scientific description of material in study, you have to add the information on the mass of LFP per 1 cm2 of electrode and the amount of carbon per 1g of LFP after the coating.
3) «As a product, a green precipitate was obtained, which was washed with deionized water, filtered and dried at 50°C. To perform carbon coating, synthesized LFP was suspended…»
- The color of LFP powder is black or grayish-black. Hence, even in the lack of XRD data it is clear that the product of solvothermal processing in this study was not LFP itself, but its precursor. In fact, the formation of carbon-coated LFP was performed during thermal treatment of this precipitate with carbon precursor in N2/H2 at 600 C. Please correct the description of the synthesis procedures in the chapter 2.1.
4) Fig. 1a. The appearance of the term “demagnetization” in this scheme looks strange. It seems that the magnetic separation process was actually applied instead in order to remove the magnetic secondary phases.
5) Taking into account the previous considerations, EIS studies don’t look really necessary; I would better move it to the Supplementary. However, as Li+ diffusion coefficient is calculated, you have to compare it with similar values obtained by other groups for LFP-based materials.
